# Impact of Caterpillar Increased Feeding Rates on Reduction of Bt Susceptibility

**DOI:** 10.3390/ijms232314856

**Published:** 2022-11-28

**Authors:** Anirudh Dhammi, Jaap B. van Krestchmar, Jiwei Zhu, Loganathan Ponnusamy, Fred Gould, Dominic Reisig, Ryan W. Kurtz, R. Michael Roe

**Affiliations:** 1Department of Entomology and Plant Pathology, College of Agriculture and Life Sciences, North Carolina State University, Raleigh, NC 27695, USA; 2Department of Entomology and Plant Pathology, Vernon G. James Research & Extension Center, Plymouth, NC 27962, USA; 3Cotton Incorporated, 6399 Weston Parkway, Cary, NC 27513, USA

**Keywords:** tobacco budworm, bollworm, fall armyworm, cotton, *Bacillus thuringiensis*, MVPII, Cry1Ac, Cry2Ab2, behavioral resistance, increased feeding

## Abstract

The use of insect-resistant transgenic crops producing *Bacillus thuringiensis* protein Cry toxins (Bt) to control caterpillars is wide-spread. Development of a mechanism to prevent Bt from reaching its target site in the digestive system could result in Bt resistance and resistance to other insecticides active *per os*. Increased feeding rates by increasing temperature in tobacco budworms, *Chloridea virescens*, and bollworms, *Helicoverpa zea*, decreased Bt Cry1Ac susceptibility and mortality. The same was found in *C. virescens* for Bollgard II plant extract containing Bt Cry1Ac and Cry2Ab2 toxins. Furthermore, *H. zea* from the same inbred laboratory colony that fed faster independent of temperature manipulation were less susceptible to Bt intoxication. A laboratory derived *C. virescens* Bt resistant strain demonstrated a higher feeding rate on non-Bt artificial diet than the parental, Bt susceptible strain. A laboratory-reared Bt resistant fall armyworm, *Spodoptera frugiperda*, strain also fed faster on non-Bt diet compared to Bt susceptible caterpillars of the same species, both originally collected from corn. The studies in toto and the literature reviewed support the hypothesis that increased feeding rate is a behavioral mechanism for reducing caterpillar susceptibility to Bt. Its possible role in resistance needs further study.

## 1. Introduction

The use of insect-resistant, transgenic crops is wide-spread [1,2,3,4,5,6]. They are safe for consumers and the environment reducing the need for chemical sprays, are highly efficacious for some pest caterpillars of significant economic importance, and simplify the farmer’s pest management practices. The insect protein toxins used in transgenic plants are derived from the bacterium *Bacillus thuringiensis* (Bt). To address resistance evolution in the field, industry is expressing multiple Bt proteins in the same cultivars [7].

Laboratory and field data demonstrated that insect pests were capable of evolving resistance to the Bt toxins [8,9,10,11,12,13,14]. For example, in a field study conducted before the commercial release of Bt crops in 1996, the diamondback moth, *Plutella xylostella* (L.) (Lepidoptera: Plutellidae), was able to develop resistance to Bt sprays [8]. Furthermore, Gould et al. [13] developed a laboratory strain (YHD2) of the tobacco budworm, *Chloridea virescens* (F.) (Lepidoptera: Noctuidae), that was highly resistant to Bt. Bt resistance in the field is now occurring for multiple insect pest species for multiple field crops. Examples include but are not limited to the fall armyworm, *Spodoptera frugiperda* (J.E. Smith) (Lepidoptera: Noctuidae), in Puerto Rico and North Carolina (USA); the maize stalk borer, *Busseola fusca* (Lepidoptera: Noctuidae), in South Africa; the pink bollworm, *Pectinophora gossypiella* (Saunders) (Lepidoptera: Gelechiidae), in India and the USA; and the bollworm, *Helicoverpa zea* (Boddie) (Lepidoptera: Nocutuidae), in the USA [5,6,15,16,17,18,19,20,21]. *Helicoverpa zea*, has been reported to have resistance to multiple Cry family proteins, including Cry1Ac, Cry1A.105, Cry1Ab, Cry2Ab2 and Cry1F [6,19,20,22]. Resistance to Cry1A family proteins has plateaued in *H. zea* but Cry2A family resistance is still developing [22]. Recently, in *H. zea*, resistance to Vip3Aa was detected in the USA [3,23].

There are multiple recognized mechanisms for Bt-resistance in insects. For examples, there are alterations in gut serine proteases, Bt-receptors (cadherin, alkaline phosphatase, and aminopeptidase) including transporters (ABC transporters), and tetraspanin [24,25,26,27,28,29,30]. We recently reported on the potential involvement of the insect immune system in Bt-resistance (to reduce sepsis) and increased P450 transcription (to detoxify metabolites from bacteria proliferation and insect cell lysis) [31]. Bt-receptors have been linked to resistance via mutations that alter binding of Bt-proteins to midgut receptors [32,33]. The ABCC2 gene recently was found not to be a Bt-receptor in *H. zea*; alkaline phosphatase 2 was attributed to Cry1Ac but not Cry2Ab resistance [34,35]. Yang et al. [23] reported that resistance to Vip3Aa (a non-Bt Cry toxin engineered into crop plants) was monogenic, autosomal, and recessive; however, the exact gene annotation is unknown.

Food absorption efficiency across the insect digestive system decreases with an increasing rate of food movement through the digestive tract [36]. An increased feeding rate limits the time for diffusion across the peritrophic membrane to the ventricular epithelium and for absorption across the brush border. Bailey et al. [37] showed differences in feeding rate measured by the rate of fecal production for bollworms collected from different field sites in the SE US prior to the wide-spread field use of Bt transgenic crops. They found higher feeding rates were associated with lower susceptibility to Cry1Ac toxin. Cabrera, et al. [38] found natural variations in feeding rates measured by the rate of fecal production by as much as 2.5-fold in three populations of the tobacco budworm collected as eggs from the field in three North Carolina counties. These data suggested that there were natural variations in insect feeding rates, and it was hypothesized that these differences might affect Bt toxin susceptibility and caterpillar survivorship on Bt crops. The current study was to test this hypothesis by artificially increasing the feeding rate with increased temparture in different caterpillar species and selecting for differences in feeding rates in the same caterpillar population to determine the impact on Cry susceptibility; and comparing the feeding rate between Bt susceptible and resistant populations for two caterpillar species.

## 2. Results

### 2.1. Temperature Elicited Reduced Food Consumption Increased C. virescens Bt Susceptibility

When Bt susceptible, lab strain, neonates of *C. virescens* (YDK) were reared at 30 compared to 20 °C on non-Bt artificial diet, there was a lower feeding rate (lower fecal production) at the lower temperature (Table 1; *p* ≤ 0.001). Insects are ectotherms, and their rate of metabolism and feeding are lower when the temperature is lowered. These studies were conducted using a feeding disruption assay where neonate feeding rate over 24 h is directly proportional to the rate of fecal pellet production [38,39,40,41]. Fecal production at 20 °C (an average of 27.33 pellets per neonate per 24 h) was 42.12% of the fecal production at 30 °C (64.89 per neonate, Table 1), a 2.4-fold difference.

MVPII Bt Cry1Ac toxin was added to the diet at a dose that reduced fecal production at 30 °C to 54.48% of the non-Bt control at the same temperature; this was a change from 64.89 pellets per neonate after 24 h with no Bt in the diet to 35.35 pellets per neonate with Bt in the diet at 30 °C (Table 1; *p* ≤ 0.001). Based on the observed effect of a 10 degree temperature reduction alone on fecal production in the absence of Cry1Ac in the diet, the predicted fecal production for neonates fed diet containing Bt at 20 versus 30 °C was 14.80 ± 4.86 (±1 standard error of the mean (SEM); *n* = 48) fecal pellets per neonate. However, the actual fecal production rate was 66.82% of the expected rate (Table 1, 9.95 pellets; *p* = <0.001) suggesting an increase in the insect’s susceptibility to the Bt toxin. We have shown before that Bt susceptibility can be directly measured by the rate of fecal production in 24 h in the feeding disruption assay, i.e., the greater the Bt susceptibility, the lower the rate of fecal production [38,39,40,41,42,43,44]. In the experiments just described, all of the larvae were alive after 24 h.

The impact of this same assay with Bt in the diet at the same two temperatures on larval mortality after 7 d on the Bt diet also was investigated (Table 1). A 10 °C decrease in temperature (Table 1) which decreases the feeding rate by 2.4-fold on non-Bt diet increased the percentage mortality 1.7-fold from 35.94 to 62.50% (*p* ≤ 0.001) in the feeding disruption assay with Bt toxin from MVPII in the diet. This change in incubation temperature had no effect on mortality in the absence of Bt toxin in the diet (Table 1), where percentage mortality was 3.39 and 4.69%, respectively (*p* = 0.690). Using both fecal production and mortality as end points, increased susceptibility to Bt toxin was correlated with a reduced feeding rate.

The results presented in Table 1 with MVPII and the mortality endpoint were repeated using the same experimental conditions but with an aqueous extract of Bollgard II cotton leaves as a source for Cry1Ac and Cry 2Ab2 toxin. A leaf dose was used that produced 45.83% mortality in a 7 day bioassay at 30 °C (Figure 1A). As shown in Figure 1A, a 10 °C decrease in temperature which decreased the feeding rate 2.4-fold on non Bt artificial diet, increased the percentage mortality at 20 °C to 75.00% (*p* ≤ 0.001). This change in incubation temperature had no effect on mortality in the absence of Bollgard II extract in the diet, where survival was 92.85 and 93.75%, respectively (Figure 1B; *p* = 0.798). The studies in toto argue that a reduced rate of feeding increases *C. virescens* neonate susceptibility to Bt for both MVPII Cry1Ac toxin and for Bollgard II leaf extract Cry1Ac and Cry 2Ab2 toxins.

### 2.2. Temperature Elicited Reduced Food Consumption Rate Increased H. zea Bt Susceptibility

Similar results to *C. virescens* were found for *H. zea* (Table 1). A reduction from 30 to 20 °C reduced fecal production in *H. zea* to 43.10% of that at 30 °C on non-Bt artificial diet (Table 1). When a diagnostic dose of MVPII was incorporated into the artificial diet, the fecal production rate at 30 °C was reduced from 76.75 pellets per neonate in 24 h to 34.44 pellets per neonate (Table 1; *p* ≤ 0.001), a reduction of 44.87%. The predicted change in fecal production with MVPII in the diet based on a 10 °C temperature reduction was 14.88 ± 1.57 (+SEM, *n* = 48) pellets. The observed production was 8.27 pellets per neonate (Table 1), 53.53% of what was predicted by a temperature effect alone (*p* ≤ 0.001).

We also examined the impact of temperature change using mortality as an end point for *H. zea* (Table 1). As was the case previously shown for neonates of *C. virescens*, a decrease of 10 °C for *H. zea* increased the percentage mortality of a diagnostic dose of MVPII in artificial diet from 51.70 to 73.33% in a 7 day bioassay (Table 1; *p* = 0.002). This temperature change had no impact on percentage mortality in the absence of Bt toxin in the diet (Table 1; *p* = 0.545).

### 2.3. Development of Methodology to Identify Naturally Occurring Slow and Fast Feeding H. zea Neonates

Figure 2A is the increasing number of fecal pellets in the first 12 h for individual neonates and the corresponding fecal pellets produced from the same insect in the following 24 h. The majority of the neonates could be organized into two categories, “slow feeders” generating 1–15 fecal pellets per insect in the first 12 h and “fast feeders” producing 30–50 pellets per neonate in the first 12 h (Figure 2B). When we segregated the data into slow and fast feeders using this approach and examined the differences in feeding between these two groups in the following 24 h, we found this selection method was able to separate slow and fast feeders in the following 24 h (compare gray bars in Figure 2C; *p* ≤ 0.001). This method provided an opportunity to examine the effect of feeding rate on Bt susceptibility in the same inbred *H. zea* population without having to change temperature.

### 2.4. Impact of Feeding Rate at a Constant Temperature on Susceptibility to MVPII Cry1Ac

Figure 3 shows the difference between fecal production on non-Bt artificial diet versus fecal production when a diagnostic dose of MVPII was added to the diet for slow versus fast feeding *H. zea* allowed to feed for 24 h at 25 °C. The fold difference between diet without and with Bt for slow feeders was 7.87 and 3.78 for fast feeders (*p* = 0.014 and *p* = 0.003, respectively). The impact of the MVPII in the diet on fecal production was less for the fast feeders by 2.1-fold (7.87/3.78). Bt susceptibility in these experiments were measured by the relative reduction in fecal pellet production between no Bt versus Bt in the diet. Slow feeders were more susceptible to Bt poisoning than fast feeders at the same temperature. The outcome was consistent with earlier results presented where reduced feeding rate elicited by a lower temperature increased Bt susceptibility measured by both fecal production and mortality for *C. virescens* and *H. zea* (Table 1, Figure 1).

### 2.5. Differences in Feeding Rates between Bt Susceptible and Bt Resistant Caterpillar Strains on Non-Bt Artificial Diet

We examined the feeding rate on non-Bt artificial diet (measured by the rate of fecal production) in neonates of a Bt-susceptible laboratory strain of *C. virescens* (YDK strain) compared to a Bt-resistant laboratory strain of the same species (YHD2 was derived from YDK by Bt toxin selection) at the same test temperature and using the feeding disruption assay. The feeding rate for the resistant strain (34.64 pellets per insect in 24 h) was 1.33-fold greater than for the susceptible strain (26.14 pellets per 24 h; *p* = 0.016; Figure 4).

The same comparison was conducted on a Bt resistant *S. frugiperda* laboratory strain established from the field in North Carolina from Herculex field corn to a susceptible laboratory strain of the same species established from the field in multiple SE USA locations from corn. Both strains were reared by the same methods in the same laboratory at NC State University. These studies also used the feeding disruption assay method. We found that the resistant strain on non-Bt artificial diet fed 3.37 times faster (38.97 pellets per 24 h) than the susceptible strain (11.55 pellets per 24 h; *p* ≤ 0.001; Figure 5).

## 3. Discussion

Bailey et al. [37] established and maintained laboratory colonies of *H. zea* on artificial diet that were collected from different field sites in the SE USA. They were maintained in the laboratory only long enough to measure their feeding rates and susceptibility to Cry1Ac toxin. The feeding rates measured by the rate of fecal pellet production was different between these different colonies with higher feeding rates correlated with lower susceptibility to Cry1Ac toxin. Cabrera et al. [38] collected *C. virescens* eggs from insecticide untreated tobacco in three North Carolina counties. When these eggs were separated from the tobacco in the laboratory, allowed to hatch, and the neonates placed on artificial non-Bt diet, fecal pellet production rate (a measure of food consumption) varied as much as 2.5-fold between locations. Bailey et al. [37] also found the feeding rate for *C. virescens* for field strains from the SE US were much lower than that of *H. zea* with the lower feeding rate of *C. virescens* correlated with greater susceptibility to Cry1Ac than the bollworm. It was hypothesized from this research, that differences in feeding rates could be a natural mechanism for Bt tolerance, a possible mechanism for Bt resistance, and a possible mechanism for cross-resistance between Bt and other protein or large molecular weight insecticides that might be developed in the future.

To test this hypothesis further in this current study, changes in temperature were used to artificially alter the feeding rate of *C. virescens* and *H. zea* Bt-susceptible neonates from highly inbreed laboratory colonies. The colonies were maintained on artificial diet, and feeding rate was measured by the rate of fecal pellet production using a feeding disruption assay. Caterpillar feeding rate, Bt and chemical insecticide susceptibility, and Bt and chemical insecticide resistance was shown before in several studies could be measured by the rate of fecal pellet production in neonate caterpillars after a 24 h incubation period using the feeding disruption assay [38,39,40,41,43,44]. When a diagnostic dose of insecticide was added to the hydratable meal pad in this assay for example, insecticide susceptibility was proportional to the rate of fecal pellet production, i.e., the greater the Bt susceptibility, the lower the number of fecal pellets. Fecal pellet production rate is also proportionally to toxin dose with this assay. The insects were isolated in a specialized 16-well plate in these experiments.

In the research reported here, a 10 °C reduction in temperature (from 30 to 20 °C) reduced the feeding rate in the absence of MVPII Cry1Ac Bt toxin in the artificial diet. When a diagnostic dose of MVPII Cry1Ac toxin was added to the diet, fecal production at 30 °C was reduced to about one-half. However, the actual reduction in fecal production observed at 20 °C with Bt in the diet was greater than what was predicted from the temperature reduction alone, for both *C. virescens* and *H. zea*. When these experiments were repeated for both species using 7 d mortality as the endpoint, the percentage mortality was greater at 20 than 30 °C. Temperature changed had no effect on mortality in the absence of Bt toxin in the diet. Similar results were obtained when MVPII was replaced with a Bollgard II water extract in the artificial diet. Bollgard II produces both Cry1Ac and Cry2Ab2 Bt toxin. These results were counter intuitive because (i) larvae at the lower temperature consumed significantly less diet and Bt toxin; (ii) any activation of the toxin by gut enzymes would be lower at 20 °C; and (iii) the rate of diffusion of protein toxin from the ingested food to Bt receptors in the gut lumen would be slower. However, the Bt toxins tested from MVPII and Bolgard II were more active at the lower temperature and lower feeding rate.

To exclude possible temperature effects on Bt susceptibility unrelated to temperature affecting the feeding rate, methods were developed to identify *H. zea* neonates (from a highly inbreed, laboratory colony) that were slow and fast feeders at one temperature. Using fecal pellet production rates as a measure of Bt susceptibility at a constant temperature, we found that the slow feeders were more susceptible to MVPII Cry1Ac toxin than fast feeders. Apparently, the feeding rate impact on Bt susceptibility was not temperature dependent. By two different methods used to change feeding rates, we found that reduced feeding rates were associated with increased susceptibility to Bt toxin.

Increased feeding rates were also found in laboratory, Bt resistant caterpillars. When a Bt resistant *C. virescens* strain was compared to its Bt susceptible parent strain, the Bt resistant strain had a higher feeding rate. This was also found for a Bt resistant *S. frugiperda* strain compared to a susceptible strain of the same species both collected from corn, where the difference in feeding rates was 3.37-fold. These results suggest that increased feeding is a possible mechanism for Bt resistance in caterpillars. However, there are other explanations for these results. For example, the higher levels of feeding in the resistant strain could be just a coincidence resulting from strain differences unrelated to Bt resistance. In the case of *C. virescens*, the resistant caterpillars (the YHD2 strain) were derived from the susceptible YDK strain and both maintained in the same laboratory under the same rearing conditions for many generations. However, maybe this selection process independent to Bt resistance also selected for an increase feeding rate. The Bt resistant and susceptible *S. frugiperda* strains were both derived from corn and reared in the same laboratory but were collected in the field from different geographical areas of the SE USA. This might explain differences in feeding rates related to the different geographical origins of the insects. In toto, our work so far is consistent with the hypothesis that feeding rate affects Bt toxin Cry susceptibility in caterpillars and suggest feeding rates might be a mechanism of Bt resistance that is not confirmed.

The mechanism for decreased Bt susceptibility at the higher feeding rate is likely explained by less Bt toxin reaching the midgut epithelium and binding to Bt midgut receptors. This is occurring both in the 24 h feeding disruption assay and in the 7 day mortality studies using Cry1Ac toxin from MVPII and for Cry1Ac and Cry2Ab2 from Bollgard II. The exact mechanism for this reduction is less clear. The food in the midgut extending to the end of the hindgut is inside a peritrophic membrane (PM), and both are moving through the digestive track at a higher rate when the feeding rate is higher. The most plausible explanations are that even though the insect is consuming more toxin, the time in the midgut at any one location is not long enough for (i) toxin solublization and/or activation, (ii) diffusion of the toxin across the PM, and/or (iii) toxin retention in the PM-ventricular epithelial space before moving into the anterior hind gut. The pores in the PM are in the 0.2 micron range. Factors that would affect movement across the PM would include toxin association with insoluble food material, enzyme processing, diffusion of protein through PM pores, and potential affinity with the PM. The function of the PM is to promote the diffusion of small molecules from food digestion like free amino acids and glucose, and the retention of undigestable food materials and biopolymers. A higher rate of PM and food movement in the midgut might also be reducing the efficacy of the counter current system in the PM-ventricular epithelial space and reducing Bt retention in the midgut in this space.

The relative importance of feeding rates versus other mechanism on Bt tolerance and resistance evolution is unknown relative to other known mechanisms. If nothing else, increased feeding might provide an additional approach by which caterpillars survive pyramided cotton technologies at a higher level, increasing their chance for the evolution of other resistance mechanisms of more significance. In the worst case, feeding rates could be a method for broad high cross-resistance to protein toxins in general and/or produce insects that feed faster and potentially cause more plant damage than the susceptible strain on non-Bt plants. The increased tolerance of caterpillars to Bt at higher temperatures is also interesting. Increased temperatures due to climate change or simply a hot summer could increase insect survival in the field on Bt crops. Differences in temperature within the cotton canopy also might affect insect positioning on the plant. How increased feeding affects the efficacy of contact insecticides is not well studied. Finally, depending on the mechanism linking feeding rate to Bt protein toxicity, increased feeding could be a mechanism for tolerance to other protein or large molecular weight insecticides.

## 4. Materials and Methods

### 4.1. Insects

Bt resistant (YHD2-strain) and susceptible (YDK-strain) tobacco budworm, *Chloridea virescens*, eggs were obtained from colonies maintained on artificial diet at North Carolina State University (NCSU) and described before [13]. Bt-susceptible bollworms, *Helicoverpa zea*, eggs were acquired from Benzon research (Benzon Research, Carlisle, PA, USA). Bt susceptible and resistant fall armyworms, *Spodoptera frugiperda*, were from colonies maintained at the Vernon G. James Research and Extension Center, NCSU (Plymouth, NC, USA) and described before [16,45]. The susceptible strain was collected from Nbt Maize (DKC 61-22 (Monsanto, St. Louis, MO, USA) and N78N-GT (Syngenta, Minnetonka, MN, USA) in Franklin Parish (LA, USA), Hendry County (FL, USA) and Hidalgo County (TX, USA) through 2008, 2011 and 2013, respectively. The Bt resistant fall armyworm population was NC-Bt-2013 collected from Herculex (Bt) corn (Corteva, Indianapolis, IN, USA) in Hyde County (NC, USA) in 2013. Both populations of the fall armyworm, were maintained on artificial diet (WARD’S Stonefly *Heliothis* diet, Rochester, NY, USA) as described by Yang et al. [46]. The resistance ratio was 85.4-fold using Bt corn leaf material [16,46]. All caterpillar eggs in this study were maintained at 25 °C, and newly emerged (in less than 24 h) neonates used for bioassays.

### 4.2. Feeding Rate and Bt Susceptibility in Caterpillars Reared at Different Temperatures

Bt susceptible YDK *C. virescens* and Benzon Bt susceptible *H. zea* neonates were bioassayed on artificial diet using hydratable meal pads and a feeding disruption assay described before in several publications [38,39,40,41,42,43,44]. In brief, the assay is 100 µL of freeze-dried, artificial caterpillar diet where the top surface of the diet is positioned flush to the bottom center of each well in a custom 16 well plate. When the meal pad is hydrated and the caterpillars feed (one per well), the plate is designed so their blue fecal pellets are deposited on the white background of the well bottom surrounding the meal pad. Blue fecal pellets on the white background facilitates easy counting of the pellets under low magnification. The diet contains a dye that does not affect caterpillar feeding. The feces is blue, indicating it originated from feeding on the hydrated meal pad. The number of fecal pellets is an index of the amount of food eaten by the caterpillar, i.e., the more eaten, the more fecal pellets produced.

The meal pads were hydrated with distilled water with a diagnostic dose of MVPII (Mycogen, San Diego, CA, USA) Cry1Ac Bt toxin (10 µg/mL in distilled water for *C. virescens* and 25 µg/mL for *H. zea*) or with a diagnostic dose of Bollgard II plant extract in distilled water for *C. virescens* (0.7 µg/mL of Bollgard II (Monsanto, St. Louis, MO, USA) plant extract containing Cry1Ac and Cry2Ab2 [38]). MVPII is a bioinsecticide containing Cry1Ac delta endotoxin [47]. When a single diagnostic dose of Bt toxin is added to the meal pad at the time of hydration, a reduction in the number of fecal pellets produced compared to that for a no Bt control meal pad provides a quantitative measure of Bt susceptibility as shown before [38,39,40,41,42,43,44]. Bt is a gut poison, and the level of poisoning affects the rate of food consumption and feces production.

Insects were allowed to feed for 24 h with no Bt in the diet after which fecal pellets were counted as a measure of the rate of food consumption. Comparing fecal production at 30 and 20 °C, a decimal percentage reduction in fecal production was calculated based on the 10 °C temperature reduction (number of fecal pellets at 20/30 °C). It was hypothesized that this same decimal percentage reduction would occur if there was Cry1Ac toxin in the diet at a diagnostic dose that would reduce but not stop feeding during the 24 h bioassay [48]. Based on this hypothesis, an expected reduction in fecal production from 30 to 20 °C was calculated from fecal production levels at 30 °C for neonates on Bt diet. The expected rate was compared to the observed fecal production rate (with Bt in the diet) at 20 °C to determine if temperature induced changes in feeding rate affected Bt susceptibility. For example, if the observed feeding rate with Cry1Ac toxin in the diet was lower than the expected at 20 °C, this would suggest that the caterpillars were more susceptible to Bt when feeding rate was reduced by temperature.

Mortality assays also were conducted after 7 d at 30 versus 20 °C for the same diagnostic doses of MVPII in the feeding disruption test. Death was defined as no movement when the larvae were touched with a blunt probe. Therefore, we were able to use two different endpoints to assess the impact of temperature-elicited changes in feeding rates on Bt susceptibility. All assays were conducted with a 14:10 L:D cycle regardless of the assay end point used.

### 4.3. Bollworm MVPII Cry1Ac Bt Toxin Susceptibility in Slow and Fast Feeders

Fecal production in individual Bt susceptible *H. zea* neonates was measured at 25 °C using the feeding disruption assay on non-Bt diet for 12 h (14:10 L:D) and compared to fecal production for the next 24 h (14:10 L:D) on non-Bt diet. From these studies, we were able to develop a method using feces production in individual neonates in the first 12 h to predict slow and fast feeders during the next 24 h. Insects were sorted into two groups, slow versus fast feeders. Fecal production was then measured without and with the diagnostic dose of MVPII Cry1Ac Bt toxin in the diet used before for *C. virescens* using the feeding disruption assay (described earlier) at 25 °C for slow and fast feeders over a 24 h period (14:10 L:D).

### 4.4. Fecal Production Rate for Bt Susceptible versus Resistant Caterpillars

Fecal production rates were measured for neonates of susceptible and resistant *C. virescens* [13] and *S. frugiperda* [16] (one larva per well) on non-Bt diet by methods described in Section 4.2. The assays were conducted at 25 °C for 24 h at 14:10 L:D, after which the number of fecal pellets were counted as an index of the rate of feeding.

### 4.5. Data Analysis

Normality of each data set was tested using the Shapiro–Wilk Normality Test. When distributions were not shown to be normal, data were log_10_ (x + 1) transformed to normalize the distribution. The student’s *t*-test was used to test for significant differences with alpha = 0.05. Statistical analyses were performed using SigmaPlot (version 14, Systat Software, Palo Alto, CA, USA).

## Figures and Tables

**Figure 1 ijms-23-14856-f001:**
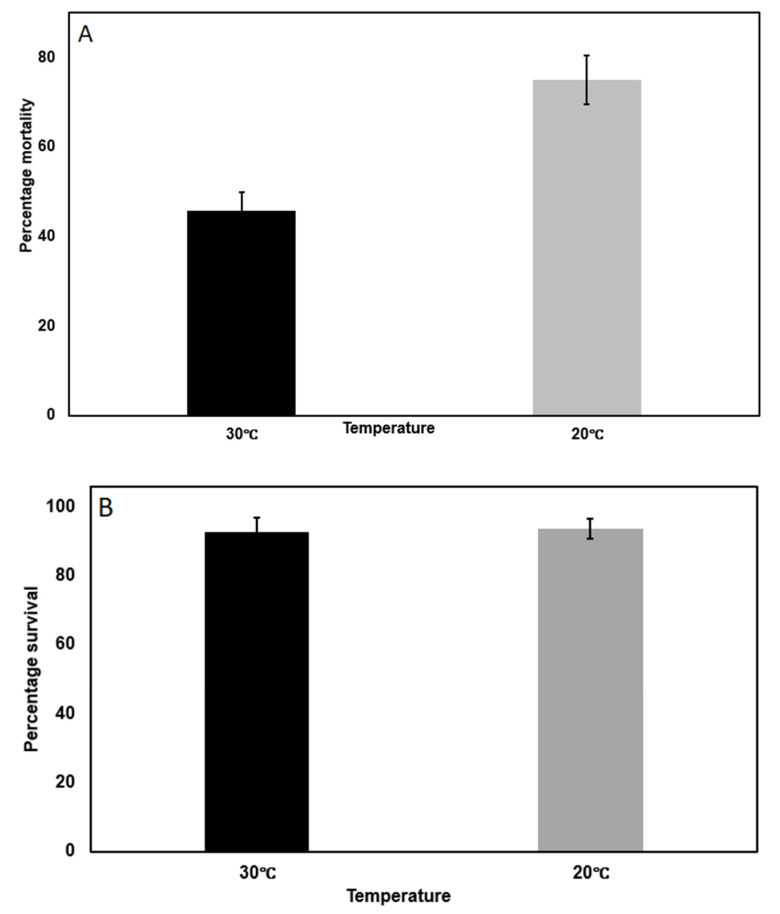
(**A**) Impact of temperature on mortality for Cry1Ac-susceptible *Chloridea virescens* neonates from a standard lab (YDK) strain (n = 48 for each treatment) on artificial diet containing a diagnostic dose of Bollgard II cotton leaf extract (in the feeding disruption assay). (**B**) Impact of temperature on survival for Cry1Ac-susceptible *Chloridea virescens* neonates (YDK strain) containing conventional cotton leaf extract (with no Bt toxin in the feeding disruption assay; n = 48 for each treatment)). The insects were maintained on the test diet for 7 d at the temperatures shown using the feeding disruption assay. Mortality was defined as no movement when the insect was touched with a blunt probe. Error bars are ±1 standard error of the mean.

**Figure 2 ijms-23-14856-f002:**
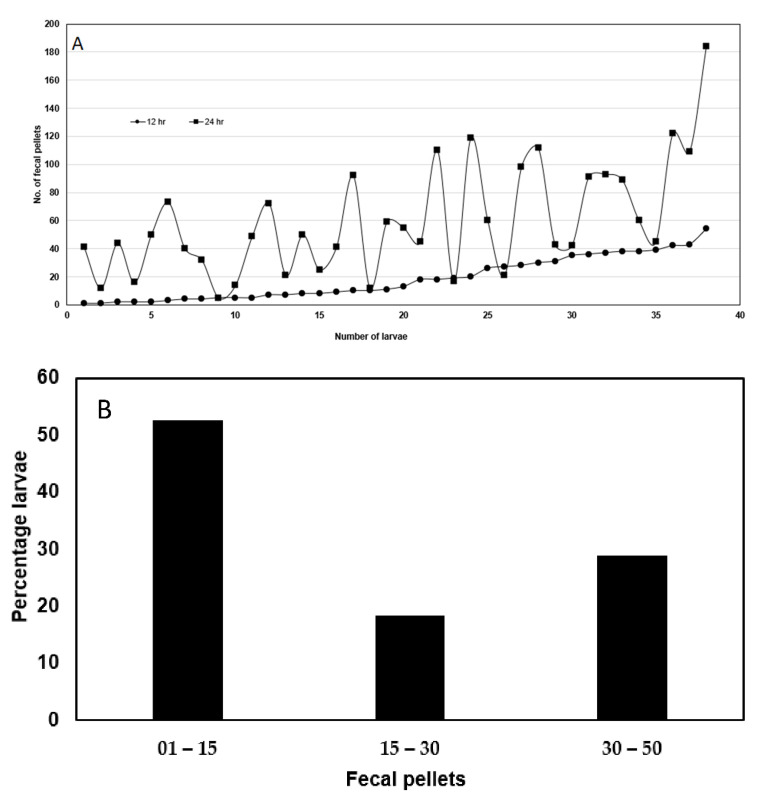
(**A**) Fecal production rate per neonate at 25 °C during their first 12 h as a larva versus their feeding rate during the next 24 h for Cry1Ac-susceptible *Helicoverpa zea* from a standard lab strain on artificial diet without Bt (in the feeding disruption assay). The data were organized on the X-axis based on increasing number of fecal pellets produced in the first 12 h for individual neonates and the corresponding fecal pellets produced from the same insect in the following 24 h plotted directly above (n = 38). (**B**) Percentage of larvae in (**A**) that were slow feeders (produced 1–15 fecal pellets) versus fast feeders (produced 30–50 fecal pellets) in the first 12 h. (**C**) The rate of feeding based on the number of fecal pellets produced in the first 12 h was used to sort larvae as slow and fast feeders and successfully predicted slow and fast feeders over the next 24 h. Error bars are ±1 standard error of the mean.

**Figure 3 ijms-23-14856-f003:**
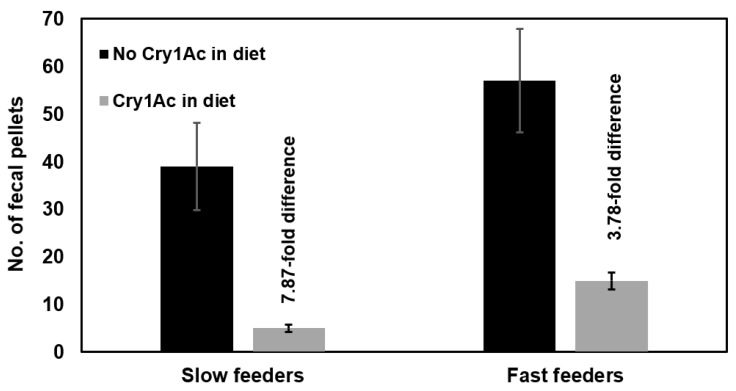
Fecal production for Cry1Ac-susceptible *Helicoverpa zea* neonates (slow versus fast feeders, see Figure 2) over a 24 h incubation period on artificial diet without Bt versus with a diagnostic dose of MVPII in the diet (using the feeding disruption assay). Slow feeders demonstrated an increased susceptibility to MVPII. Error bars are ±1 standard error of the mean (n =20 for each treatment).

**Figure 4 ijms-23-14856-f004:**
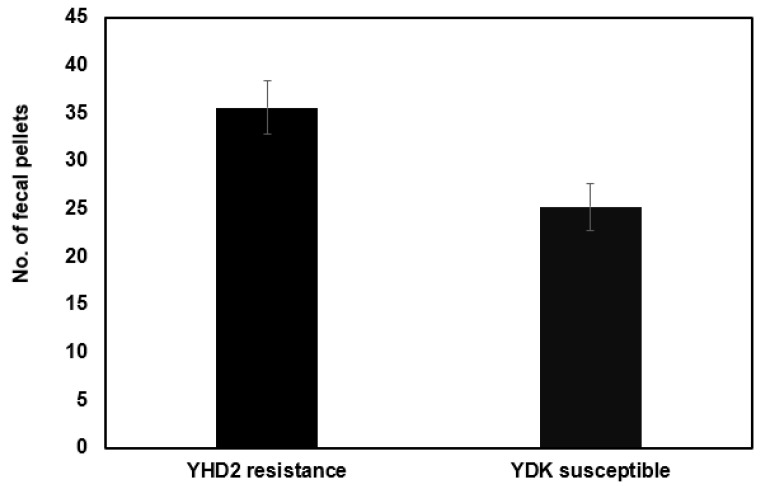
Fecal production over 24 h of Cry1Ac-susceptible *Chloridea virescens* neonates (YDK strain) versus Cry1Ac resistant *C. virescens* neonates (YHD2 strain) on artificial diet containing no Bt toxins (using the feeding disruption assay) (n = 56 and 64, respectively). Error bars are ±1 standard error of the mean.

**Figure 5 ijms-23-14856-f005:**
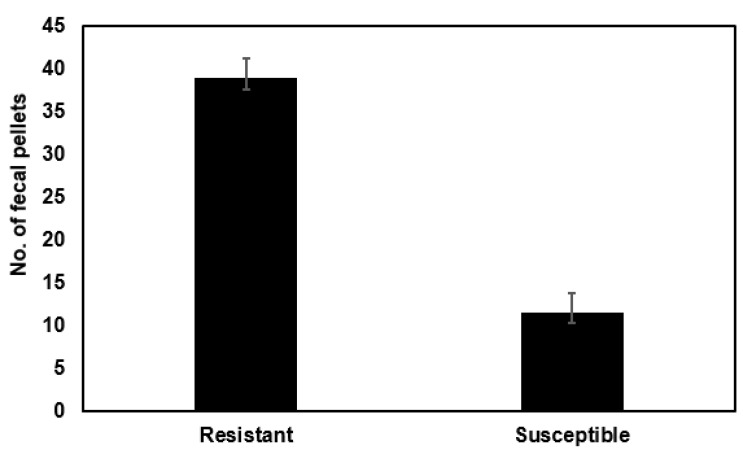
Fecal production over 24 h of Cry1Ac-susceptible *Spodoptera frugiperda* neonates versus Cry1Ac resistant *S. frugiperda* neonates on artificial diet containing no Bt toxins (using the feeding disruption assay; n = 80 for each treatment). Error bars are ±1 standard error of the mean.

**Table 1 ijms-23-14856-t001:** Variation in fecal production after 24 h and percentage mortality after 7 days for Cry1Ac-susceptible *Chloridea virescens* and *Helicoverpa zea* neonates on artificial diet (without and with MVPII-Cry1Ac toxin in the diet) at 30 and 20 °C.

Insect	Treatment	Number of Fecal Pellets ± SEM (n) ^a^		Percentage Mortality ± SEM (n) ^b^	
		30 °C	20 °C	30 °C	20 °C
*C. virescens*	No Bt	64.89 ± 4.25 (48)A	27.33 ± 2.06 (48)B	3.39 ± 2.04 (8)a	4.69 ± 2.28 (8)a
	MVPII	35.35 ± 3.73 (48)A	9.95 ± 1.36 (48)B	35.94 ± 3.68 (8)a	62.50 ± 3.34 (8)b
*H. zea*	No Bt	76.75 ± 4.91 (48)A	33.08 ± 2.12 (48)B	3.33 ± 3.33 (5)a	6.60 ± 4.03 (5)a
	MVPII	34.44 ± 2.78 (48)A	8.27 ± 0.59 (48)B	52.33 ± 3.33 (5)a	73.33 ± 3.12 (5)b

^a^ Different letters (A,B) on the same line for fecal pellets results indicate statistically significant differences based on a *t*-test (alpha = 0.05) where n is the number of individual insects tested. SEM = standard error of the mean. ^b^ Different letters on the same line (a,b) for percentage mortality indicate statistically significant differences based on a *t*-test (alpha = 0.05) where n is the number of replicates where each replicate was 16 insects. SEM = standard error of the mean.

## Data Availability

Not applicable.

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
