# Peer review of "Impact of Caterpillar Increased Feeding Rates on Reduction of Bt Susceptibility"

_ijms, 2022, doi:10.3390/ijms232314856_

Round 1
Reviewer 1 Report
This manuscript offers compelling evidence that increased feeding rates in different caterpillar species can reduce susceptibility to the Bt toxins Cry1Ac and Cry2Abc. The proposed mechanism of reduced transport of toxin across the peritrophic membrane based on speed of digestion is consistent with the results. I found the data presented to be convincing and of interest as a novel mechanism of resistance. I think the manuscript is ready for publication as it is, and I find myself in the relatively unique position of not having any suggested changes (well, other than changing the word "maintain" on line 276 to "maintained". Before closing, I will say that I disagree with the authors statements in the introduction, regarding whether or not Bt-engineered plants are an IPM tactic. From my perspective, the Bt-plants are essentially insecticide delivery systems, and represent prophylactic use of insecticide which is very much in opposition to the notion of use of intervention only as necessary, which is at the core of IPM philosophies. However, I offer this only as a point of disagreement/discussion and not as an issue I think the authors need to address in a revision (basically, reasonable people can disagree on this point).
Author Response
Reviewer 1
This manuscript offers compelling evidence that increased feeding rates in different caterpillar species can reduce susceptibility to the Bt toxins Cry1Ac and Cry2Abc. The proposed mechanism of reduced transport of toxin across the peritrophic membrane based on speed of digestion is consistent with the results. I found the data presented to be convincing and of interest as a novel mechanism of resistance. I think the manuscript is ready for publication as it is, and I find myself in the relatively unique position of not having any suggested changes (well, other than changing the word "maintain" on line 276 to "maintained". Before closing, I will say that I disagree with the authors statements in the introduction, regarding whether or not Bt-engineered plants are an IPM tactic. From my perspective, the Bt-plants are essentially insecticide delivery systems, and represent prophylactic use of insecticide which is very much in opposition to the notion of use of intervention only as necessary, which is at the core of IPM philosophies. However, I offer this only as a point of disagreement/discussion and not as an issue I think the authors need to address in a revision (basically, reasonable people can disagree on this point).
Authors’ response: Thank you for the positive assessment of our paper. We agree the statement made about IPM in the introduction is complicated and can be “looked-at” in different ways. The view of the reviewer is valid and thoughtful. Since the statement was not critical to the research presented, we deleted the statement about IPM from the introduction.
“Maintain” was changed to “maintained”.
Reviewer 2 Report
The manuscrip entitled: Impact of Caterpillar Increased Feeding Rates on Reduction
of Cry1Ac Susceptibility and Possible Resistance to Bt Crops, by Anirudh Dhammi,
Jaap B Van Krestchmar, Jiwei Zhu, Loganathan Ponnusamy, Fred Gould , Dominic
Reisig, Ryan W. Kurtz, Richard Michael Roe has merit and I point out some aspects
that I consider relevant..
1. Title too long and confusing, shorten it to make it more attractive;
2.The abstract feels a little too long and hard to follow along. Using abbreviations could help to improve readability. The authors could be more direct regarding to most important the results;
3. In Keywords: Bt and Cry1Ac are already found in the title, to replace;
4. The introduction is very clear and well written. The hypothesis stablished but there is no mention about how the authors are going to test the hypothesis. This could be added;
5. Line 41 - Complete scientific name information is needed when it appears the first time. This should be corrected throughout the text;
6. line 52-55 - I am not sure if this part is needed or well placed;
7. Line 67-68 - this part could be deleted for better readability;
8. In results, I suggest the authors to be more direct, focusing and clarifying the most important results. That would increase the readability of the text;
9. line 89-96 - That’s part of the Material and methods;
10. The table 1 must be formatted properly;
11. Figure 1 - it seems redundant to evaluate mortality and survival in the same conditions;
12. Line 195-197 - that’s already explained in the Material and methods;
13. The discussion lists and describes well all the processes carried out in the work, discussing them along the text;
14. The methodology brings some results and discussion to the text, that should be reviewed. At the same time the addition of images could help to better understand the methodological processes;
15. Line 422 - 423 ...a decimal percentage reduction in fecal production was calculated based on the 10 ℃ temperature reduction...how was it calculated;
16. Line 423-424 - ....Temperature reduces feeding and fecal production because the lower temperature lowers the metabolism and activity of the insect....This should not be in Material and methods;
17. Line 424-427 .....It was hypothesized that this same decimal percentage reduction would occur if there was Cry1Ac toxin in the diet at a diagnostic dose that would reduce but not stop feeding during the 24 h bioassay period.....Is there any reference that could reinforce this hypothesis?
18. Line 427-429 - ...Based on this hypothesis, an expected reduction in fecal production from 30 to 20 ℃ was calculated from fecal production levels at 30 ℃ for neonates on Bt diet......Is there any reference that could reinforce this hypothesis?
19. Line 455 - Fecal production rates were measured, how were they measured? It could be clearer.
Author Response
Reviewer 2
The manuscrip entitled: Impact of Caterpillar Increased Feeding Rates on Reduction
of Cry1Ac Susceptibility and Possible Resistance to Bt Crops, by Anirudh Dhammi,
Jaap B Van Krestchmar, Jiwei Zhu, Loganathan Ponnusamy, Fred Gould , Dominic
Reisig, Ryan W. Kurtz, Richard Michael Roe has merit and I point out some aspects
that I consider relevant..
- Title too long and confusing, shorten it to make it more attractive;
Authors’ response: Title was shorten as requested.
2.The abstract feels a little too long and hard to follow along. Using abbreviations could help to improve readability. The authors could be more direct regarding to most important the results;
Authors’ response: We significantly reduced the length of the abstract and made better use of abbreviations. Thanks for the suggestion.
- In Keywords: Bt and Cry1Acare already found in the title, to replace;
Authors’ response: Corrections made to reduce replication. Again thanks for the suggestion.
- The introduction is very clear and well written. The hypothesis stablished but there is no mention about how the authors are going to test the hypothesis. This could be added;
Authors’ response: This was added at the end of the introduction.
- Line 41 - Complete scientific name information is needed when it appears the first time. This should be corrected throughout the text;
Authors’ response: Correction made as suggested.
- line 52-55 - I am not sure if this part is needed or well placed;
Authors’ response: The sentence was removed and is not needed.
- Line 67-68 - this part could be deleted for better readability;
Authors’ response: We agree and this part was removed.
- In results, I suggest the authors to be more direct, focusing and clarifying the most important results. That would increase the readability of the text;
Authors’ response: We were not exactly sure of all aspects of the results where changes were recommended. We did edit the results to remove information as much as possible that was also described in the M&Ms. This information was originally added to the results for clarification since in the IJMS, the M&Ms follow the Results section. We thought the information was critical in understanding the results.
We also made changes suggested in additional specific comments made by the reviewer that follows relative to comments on the Results section.
- line 89-96 - That’s part of the Material and methods;
Authors’ response: We removed about 30% of the text leaving only what is the minimum needed to understand how feeding rate was measured indirectly by the rate of feces production and the reason temperature affects feeding rate in insects.
- The table 1 must be formatted properly;
Authors’ response: Corrections were made as suggested.
- Figure 1 - it seems redundant to evaluate mortality and survival in the same conditions;
Authors’ response: Mortality (Fig. 1A) was measured in the presence of Bt cotton leaf extract in the diet. Survival (Fig. 1B) was measured in the presence of non-Bt cotton leaf extract in the diet. This was explicit in the figure caption and in the Results describing the findings shown by Figure 1.
The point of the comparison was to show that cotton leaf extract alone when no Bt does not kill the neonates. We used survival instead of mortality in Fig. 1B because the low mortality when plotted with error bars would have been close to zero.
- Line 195-197 - that’s already explained in the Material and methods;
Authors’ response: The lines of text were removed as suggested.
- The discussion lists and describes well all the processes carried out in the work, discussing them along the text;
Authors’ response: Thanks for the complement.
- The methodology brings some results and discussion to the text, that should be reviewed. At the same time the addition of images could help to better understand the methodological processes;
Authors’ response: The authors assume this comment on adding images is related to the feeding disruption assay kit and the details on how the kit is used to measure food consumption rate by counting blue fecal pellets. This technique was described with images in multiple papers and issued US Patents that were cited. The technique also was described in detail in the M&Ms for this paper being submitted to the IJMS, but without illustrations.
We did not add figures as suggested by this reviewer since the first reviewer did not request the same and because the technique is described in detail in several, peer-reviewed publications and in US patents. However, we would be happy to seek permissions from other journals where we have published before to republish their figures in this paper under review. Please inform on how to proceed.
We have edited the methodology to remove results and discussion when possible that were unnecessary to understand the methods used.
- Line 422 - 423 ...a decimal percentage reduction in fecal production was calculated based on the 10 ℃ temperature reduction...how was it calculated;
Authors’ response: The calculation was added to the description as suggested.
- Line 423-424 - ....Temperature reduces feeding and fecal production because the lower temperature lowers the metabolism and activity of the insect....This should not be in Material and methods;
Authors’ response: This was removed as suggested.
- Line 424-427 .....It was hypothesized that this same decimal percentage reduction would occur if there was Cry1Ac toxin in the diet at a diagnostic dose that would reduce but not stop feeding during the 24 h bioassay period.....Is there any reference that could reinforce this hypothesis?
Authors’ response: A reference was added as suggested. This was a hypothesis based on the information in the reference. There is no reason to expect the Q10 for food consumption would be affected by the presence of Bt or any other materials in the diet as long as they were in the diet at both temperatures investigated. Additional information follows on this point.
- Line 427-429 - ...Based on this hypothesis, an expected reduction in fecal production from 30 to 20 ℃ was calculated from fecal production levels at 30 ℃ for neonates on Bt diet......Is there any reference that could reinforce this hypothesis?
Authors’ response: We added a reference as suggested.
As it “turns out”, our hypothesis was correct since both measuring susceptibility by the level of fecal production and by mortality were in agreement in the studies reported in this paper, and these findings were furthermore validated when we compared slow versus fast feeders at the same temperature.
- Line 455 - Fecalproductionrates were measured, how were they measured? It could be clearer.
Authors’ response: We added a reference as suggested.